# Dissipative Particle Dynamics Simulation of the Sensitive Anchoring Behavior of Smectic Liquid Crystals at Aqueous Phase

**DOI:** 10.3390/molecules27217433

**Published:** 2022-11-01

**Authors:** Shiwei Chen, Jinliang Zhang, Huilong Liu, Tongyue Qiu, Haoxiang Tang, Zunmin Zhang

**Affiliations:** State Key Laboratory of Materials-Oriented Chemical Engineering, College of Chemical Engineering, Nanjing Tech University, Nanjing 211816, China

**Keywords:** dissipative particle dynamics, anchoring transition, smectic phase, nematic phase, aqueous-liquid crystal interface

## Abstract

Rational design of thermotropic liquid crystal (LC)-based sensors utilizing different mesophases holds great promise to open up novel detection modalities for various chemical and biological applications. In this context, we present a dissipative particle dynamics study to explore the unique anchoring behavior of nematic and smectic LCs at amphiphile-laden aqueous-LC interface. By increasing the surface coverage of amphiphiles, two distinct anchoring sequences, a continuous planar-tilted-homeotropic transition and a discontinuous planar-to-homeotropic transition, can be observed for the nematic and smectic LCs, respectively. More importantly, the latter occurs at a much lower surface coverage of amphiphiles, demonstrating an outstanding sensitivity for the smectic-based sensors. The dynamics of reorientation further reveals that the formation of homeotropic smectic anchoring is mainly governed by the synchronous growth of smectic layers through the LCs, which is significantly different from the mechanism of interface-to-bulk ordering propagation in nematic anchoring. Furthermore, the smectic LCs have also been proven to possess a potential selectivity in response to a subtle change in the chain rigidity of amphiphiles. These simulation findings are promising and would be valuable for the development of novel smectic-based sensors.

## 1. Introduction

In recent years, intense research efforts have been directed to the development and applications of thermotropic liquid crystal (LC)-based sensors, especially for systems involving aqueous-LC interface (thin films or droplets) [1,2,3,4]. As first proposed by Abbott et al. [5], it is essentially designed on the basis of an unique combination of the sensitive alignment responsiveness to interfaces and orientation-dependent optical texture of LCs as well as the fluid feature of aqueous-LC interfaces. As a consequence, the interfacial phenomena such as the adsorption, assembly and reorganization of target analytes can be readily transformed and amplified into distinct optical signals via macroscopic reorientation of LCs in real-time, which provides a promising platform to design simple, rapid and label-free sensors for the detection of chemical and biological species.

Numerous applications have demonstrated the effectiveness, versatility and significance of LC-based sensors in detecting various analytes of interest, including surfactants [6,7,8,9], lipids [10,11], polymers [12,13], proteins [14,15,16,17], nucleic acids [18,19], bacteria [20,21] and viruses [21]. It has been established that the sensing principle mainly relies on the spontaneous formation or reorganization of the amphiphile monolayer at the interface and the associated ordering transition in LCs. For example, a series of studies from the group of Abbott [22] have shown that the interdigitation coupling between amphiphile tails and mesogens plays an important role in triggering the planar-to-homeotropic orientation transition of LCs, and therefore the orientation, conformation and self-assembly of amphiphilic molecules are crucial. Very recently, a different mechanism was proposed that LC reorientation could be induced through lowering the orientation-dependent surface energy of LCs due to the formation of a thin isotropic layer at the aqueous interface [23,24]. While on the amphiphile-decorated LC surfaces, a reverse orientation transition of homeotropic-to-planar induced by the reorganization of amphiphiles can be further utilized for specific sensing of the bio-/chemical interactions between the analytes and functionalized surfaces, including competitive absorptions, protein bindings, enzymatic reactions and DNA hybridizations [3]. For instance, Xu et al. [19] developed a LC-based diagnostic kit for reliable detection of severe acute respiratory syndrome-corona virus-2 (SARS-CoV-2). It was observed that the LC ordering transitions could be induced by the absorption of single-stranded RNA of SARS-CoV-2 at aqueous-LC interfaces decorated with cationic surfactants and a complementary single-stranded DNA probe, which is highly selective and sensitive to the target nucleotide sequence.

In comparison to numerous promising applications, however, only a few works have been performed to improve the sensitivity and specificity of LC sensors via rational design of the molecular structure and mesophases of LC sensing materials [2]. Currently, 4-cyano-4′-pentylbiphenyl (5CB) is the most widely used nematic LC for the development of LC-based sensors due to its fluidity and fast response. Iglesias et al. [25] reported the use of mixtures of bent-core and rod-like mesogens to reduce the LC elastic constant of LCs in sensors, leading to a wider sensing range for a simple surfactant. Popov et al. [26] investigated the performance of LC sensors based on the smectic and cholesteric phases. It was found that the smectic phase could expand the range of sensitivity of LCs towards low and high surfactant concentration, while the periodic fingerprint textures of the cholesteric LCs depended on the kind of surfactants and could potentially be used for detection of molecular chirality. In view of a wide variety of LC phases as well as their unique anchoring behavior, there is still plenty of untapped opportunities to develop specific and practical LC-based sensors by extensively exploiting various non-nematic mesophases, especially with the aid of computer simulations.

Motivated by the improved sensitivity of smectic-based sensors [26], mesoscopic simulations using dissipative particle dynamics (DPD) method have been employed in the present work to explore unique anchoring behavior of nematic and smectic LCs at an amphiphile-laden aqueous-LC interface. With verified coarse-grained models developed for this interfacial system [27,28], the alignment transitions of rod-like mesogens have been investigated as a function of the surface coverage and tail rigidity of amphiphiles as well as the system temperature. Besides a good agreement with those experimental findings [26], our simulations not only yield valuable insights into the highly sensitive anchoring of smectic LCs, but also theoretically demonstrate a potential selectivity of smectic-based sensors in response to the chain rigidity of amphiphiles.

## 2. Simulation Method and Models

### 2.1. Dissipative Particle Dynamics

DPD is a popular mesoscopic simulation technique, widely used for modelling the mesoscale problems in soft matter and complex fluids [27,28,29,30]. In the simulation, a collection of atoms or molecules is coarse-grained into one DPD bead, and the dynamics of each bead is governed by Newton’s equation of motion. All the beads interact with each other via three non-bonded interactions within a specific cut-off radius $r_{c}$, including a conservative force $F_{ij}^{C}$, a dissipative force $F_{ij}^{D}$ and a random force $F_{ij}^{R}$. These pairwise forces are given by
(1)$$\begin{array}{l} F_{ij}^{C}=a_{ij}(1-r_{ij}/r_{c}){\hat{r}}_{ij}\\ F_{ij}^{D}=-\gamma \omega^{D}(r_{ij})({\hat{r}}_{ij}•v_{ij}){\hat{r}}_{ij}\\ F_{ij}^{R}=\sigma \omega^{R}(r_{ij})\theta_{ij}\Delta t^{-1/2}{\hat{r}}_{ij} \end{array}$$
where $v_{ij}=v_{i}-v_{j}$, $r_{ij}=r_{i}-r_{j}$, $r_{ij}=\left|r_{ij}\right|$ and ${\hat{r}}_{ij}=r_{ij}/\left|r_{ij}\right|$. *γ* is the friction coefficient, σ is the noise amplitude, $\theta_{ij}$ is a random number with zero average and unit variance, $\omega^{D}(r_{ij})$ and $\omega^{R}(r_{ij})$ are the distance-dependent weight functions for the dissipative and random forces, respectively. The conservative force is a soft repulsive interaction with $a_{ij}$ characterizing the maximum repulsion between each two interacting beads. The dissipative and random forces act as an in-built thermostat to maintain an equilibrium temperature T, which must satisfy the fluctuation-dissipation theorem given by the conditions
(2)$$\omega^{D}(r_{ij})={\left[\omega^{R}(r_{ij})\right]}^{2},\;\sigma^{2}=2\gamma k_{B}T$$
where $k_{B}$ is the Boltzmann constant. In this study, γ=2.66 is chosen and σ varies with *γ* and the temperature.

### 2.2. Models

To model the anchoring behavior at the amphiphile-laden aqueous-LC interface, a DPD system consisting of rod-like mesogens, amphiphile and water molecules is designed and further optimized according to the generic models developed in our previous studies [27,28]. As illustrated in Figure 1a, the thermotropic liquid crystal molecule is constructed as a semi-rigid rod-like chain with seven beads (M_7_), which is a popular generic model widely used in many DPD simulations of liquid crystal materials. To maintain the rod-like shape for mesogens, a harmonic spring potential is used to connect each two adjacent beads in the chains, and an additional bending potential is applied to control the chain rigidity, which are given as
(3)$$U_{bond}=0.5k_{bond}{\left(r-r_{0}\right)}^{2}$$
(4)$$U_{\theta }=0.5k_{\theta }{\left(\theta -\theta_{0}\right)}^{2}$$
where $r_{0}=2/3$ and $\theta_{0}=\pi$ are the equilibrium bond length and angle, respectively. With a strong setting of the bond constant $k_{bond}=500$ and bending constant $k_{\theta }=50$, a fairly rigid rod-like mesogen model is developed, which has been further proven to exhibit an isotropic-nematic transition at T=0.72±0.005 and a nematic-smectic-A transition at T=0.30±0.005. In comparison to our previous models [31], both the nematic and smectic-A phases are shifted to a higher temperature with wider phase ranges, enabling this model be more suitable for the exploration of rich anchoring behavior of different LC phases in this work. As for the amphiphile, it is represented by a diblock chain consisting of a seven-bead hydrophilic head (H) and a ten-bead hydrophobic tail (T), denoted as H_7_T_10_. The same harmonic potential with a weaker bonding constant $k_{bond}=100$ is used in the amphiphiles, while the same bending potential with $k_{\theta }=50$ is only applied on the hydrophobic tail to enhance the chain rigidity for the ordering coupling with mesogens. In addition, the water bead is chosen to be identical to one hydrophilic bead (H_1_) for simplicity.

Accordingly, there are three types of DPD beads (M, H and T) involved in the present study. For the same type, $a_{MM}=a_{HH}=a_{TT}=20$ is set according to $a_{ii}=75k_{B}/$, where ρ=4 is the bead number density of our simulated systems. For the different types, $a_{HM}=a_{HT}=40$ is employed to mimic the hydrophobic interactions and maintain a stable aqueous-LC interface. Since the interaction between the amphiphile tail and mesogens plays a crucial role in determining the anchoring state of LCs, $a_{TM}=15$ is carefully chosen on the basis of our previous simulation results, which is moderate not only to induce the homeotropic anchoring at certain conditions [27], but also to avoid the formation of some unexpected planar anchoring configurations [28]. 

### 2.3. Simulation Setup

All the simulations were performed in the canonical ensemble (NVT) using the large-scale atomic/molecular massively parallel simulator (LAMMPS) software package [32], and visualizations were rendered by OVITO [33]. The simulated systems were built as rectangular boxes with periodic boundary conditions applied in all three directions, wherein $L_{x}=L_{y}=23.1$ and $L_{z}=55.6\sim 57.7$ varied with the amount of amphiphiles (0~960), mesogens (8697~9862) and water beads (42,000). The surface coverage of amphiphiles, defined as $\rho_{s}=0.5N_{H_{7}T_{10}}/\left(L_{x}L_{y}\right)$, was tuned to assess the sensitivity of LC sensors at different temperature. The number of mesogens was kept constant at 9000 for the nematic-based sensor, while it was carefully varied with the surface coverage of amphiphiles for the smectic sensor in order to eliminate the dimensional mismatch between the spacing of smectic layers and the thickness of LC films. It should be pointed out that this kind of mismatch in this confined simulation system might be negligible in reality, since thousands smectic layers would be contained in the micrometer-sized LC films of sensors.

In general, most of simulations started from an initial configuration consisting of two neighbouring slabs of water and disordered LCs with amphiphiles randomly laden on two aqueous–LC interfaces (Figure 1b), and run at least 3 × 10^6^ time steps with ∆t=0.03 to achieve thermodynamic equilibrium. Longer simulations were required for the smectic anchoring at the lower temperature. Then the distribution and ordering of each component were thoroughly examined to identify the anchoring state of each equilibrium configuration. The orientational order parameter ($S_{u}$) of rod-like mesogens or hydrophobic tails of amphiphiles has been characterized in this paper by calculating the largest eigenvalue through the diagonalization of the ordering tensor, given as
(5)$$Q=\frac{1}{N}\sum_{i=1}^{N}\left[\frac{3}{2}{\left({\hat{u}}_{i}\right)}_{\alpha }{\left({\hat{u}}_{i}\right)}_{\beta }-\frac{1}{2}\delta_{\alpha \beta }\right],\alpha ,\beta =x,y,z.$$
where ${\hat{u}}_{i}$ is the unit vector of the long axis of the ith chain. Note that $S_{u}\approx 0.0$ is for a isotropic state while $S_{u}\approx 1.0$ is for a perfectly ordered state. In addition, the corresponding eigenvector is used to define their director, and a tilt angle is then measured as the angle between the director of mesogens or amphiphile tails and the normal direction of the aqueous-LC interface (i.e., z axis of simulated systems).

## 3. Results and Discussion

By varying the surface coverage of amphiphiles at different temperature, the orientation, arrangement and dynamics of the rod-like mesogens in nematic and smectic-A phases are thoroughly characterized to identify their equilibrium anchoring states at the aqueous-LC interface and further reveal the underlying mechanism of anchoring transitions.

Figure 2 presents the tilt angle of mesogens as a function of the surface coverage of amphiphiles as well as several instantaneous snapshots of typical anchoring configurations. It is interesting to observe that two distinct anchoring transition sequences are discovered at the temperature regimes of nematic and smectic-A phases, respectively, as shown in Figure 2a. In the nematic phase at T=0.6, the orientation of bulk mesogens are found to turn gradually from the tangential (~90°) to normal (~0°) direction of the aqueous-LC interface as the surface coverage of amphiphiles increases from 0.0 to 0.9, exhibiting a continuous planar-tilted-homeotropic anchoring transition. The representative configurations of planar, tilted and homeotropic anchoring can be seen in Figure 2b. This tendency is qualitatively consistent with the experimental observation, i.e., a continuous alignment variation of the nematic 5CB as a function of increasing the surfactant concentration [34]. While in the smectic-A phase at T=0.3, however, there is a discontinuous anchoring transition from planar to homeotropic, indicated by a sudden decrease of the tilt angle of mesogens from 86.7° at $\rho_{s}=0.35$ to 0.2° at $\rho_{s}=0.4$, as displayed in Figure 2a,c. Most importantly, this discontinuous transition in smectic-A phase occurs at a much lower surface coverage of amphiphiles in comparison to that in nematic phase, which theoretically demonstrates an improved sensitivity of the smectic-based sensors. This promising result is qualitatively in good agreement with the experiments of 8CB (4-cyano-4′-octylbiphenyl)-based sensors, wherein the alignment of smectic LCs can response to extremely low lipid concentration [26].

According to our previous studies [27,28], it has been proposed that the amphiphile monolayer laden at the aqueous-LC interface plays a role of the interfacial orientation filed and therefore determines the alignment of mesogens. In this context, the ordering of the hydrophobic tails of amphiphiles has then been characterized as a function of their surface coverage, shown in Figure 3. Owing to the favorable interaction, the hydrophobic tails prefer to penetrate into the mesogens to maximize their intermolecular contact. Therefore, even at extremely low surface coverage, the amphiphile tails exhibit a certain orientational ordering with a value of the order parameter around 0.6. With increasing surface coverage of amphiphiles, the tails become more condensed and ordered, and gradually orient to the interface normal direction, as indicated by Figure 3a. This phenomenon can be further proved by the bead density distribution of each component in typical anchoring configurations. As expected, the less ordered tails at the low surface coverage exhibit a unimodal distribution with a peak located near the aqueous-LC interface, as depicted in Figure 4a. At higher surface coverages (Figure 4b,c), it gradually turns to a uniform distribution with its width approximating to the length of amphiphile tails, indicating a vertical alignment for a complete interdigitation with mesogens.

In smectic-A phase, as presented in Figure 3b, the amphiphile tails at aqueous-LC interface show a higher orientational order parameter due to the lower temperature, and undergo a discontinuous orientation transition from parallel to perpendicular with the increase of the surface coverage. Before the transition, the loosely packed tails of amphiphiles in Figure 2c are too weak to induce one single homeotropic layer near the aqueous-LC interface, and have to orient in parallel with the interface to avoid disturbing the organization of planar smectic layers. When beyond a critical surface coverage, a sufficient condensed monolayer of the hydrophobic tails is organized with a perfect vertical alignment along the interface normal direction, and displays ten strong oscillations near the interface illustrated in the density profiles of Figure 5, corresponding to the ten beads of the amphiphile tails. As a result, the planar-to-homeotropic anchoring transition can be triggered by perfectly inducing the homeotropic smectic layers. As can be seen from the center of mass distribution of mesogens in Figure 5, there are seven periodic oscillations with a spacing of d ≈ 4.60 in LCs, where the layer spacing is approximately equal to the length of mesogens.

To deeply understand the mechanism of two distinct anchoring transitions in nematic and smectic-A phases respectively, the formation of different homeotropic anchoring states has been carefully examined by monitoring the orientation dynamics of mesogens and hydrophobic tails from the interface to bulk LCs. For the anchoring in nematic phase, each equilibrium state is usually obtained by a long simulation by cooling a disordered configuration (Figure 1b) to the target temperature. In Figure 6, we particularly characterize the formation of one homeotropic nematic anchoring at $\rho_{s}$ = 0.85 and T=0.6. It can be observed that the hydrophobic tails rapidly orient along the normal direction of the aqueous-LC interface, and simultaneously penetrate into the LCs. The mesogens near the interface are then quickly aligned with these tails as a result of the interdigitation, which gradually propagates into the bulk LCs and finally induces a well-organized homeotropic configuration. The propagation mechanism can be clearly demonstrated by the step-by-step ordering of mesogens from the interface to the bulk LCs, as shown in Figure 6.

As for the anchoring in smectic-A phase, it must be pointed out that all the equilibrium anchoring states are obtained by stepwise or directly cooling the corresponding nematic configurations to the smectic ones. If the LCs are kept in the smectic phase through the simulations, no reorientation could be observed to response to the variation of the amphiphile surface coverage. Similar phenomena has been reported in the experiments of [26], which was attributed to the stronger intermolecular interactions in the smectic phase. Figure 7 presents the dynamical variation of the tilt angle of mesogens during a tilted-to-homeotropic transition at $\rho_{s}$ = 0.4 by directly cooling from T=0.6 to T=0.3. It is very unique to observe that the reorientation dynamics is mainly governed by the appearance and growth of smectic layers. As displayed in Figure 7a, the mesogens rapidly become more ordered in the first 1 × 10^5^ time steps, and then begin to slowly adjust their orientation towards the interface normal direction until the appearance of smectic layers at 1.5 × 10^6^ time steps, indicated by the weak periodic oscillations along the z axis appearing in the density profile of Figure 7b. As the simulation goes on and the density peaks turn to be more regular and sharper, the reorientation to homeotropic alignment is then significantly accelerated by the quick growth of those smectic layers. In the whole process, the homeotropic smectic layers along the interface normal direction are organized and adjusted almost in a synchronous manner, which is clearly different from the propagation mechanism in the nematic phase. Note that a larger system with eleven smectic layers has also been examined, and no difference was observed except a longer simulation. However, the mismatch between the spacing of smectic layers and the thickness of LC films indeed significantly affects the alignment of smectic LCs in our simulations, which was found to distort homeotropic anchoring to be tilted or partially planar. In view of the macroscopic size (~20 µm) of LC films in the sensors, the effect of dimensional mismatch in our small-sized simulation systems would be negligible in practice.

In addition, amphiphiles with tunable tail rigidity have been further employed to explore potential sensing selectivity of nematic and smectic LCs. When decreasing the bending constant $k_{\theta }$ from 50 to 10, the hydrophobic tails of amphiphiles become more flexible and less ordered in the interfacial monolayers, which is found to weaken their ability to direct the alignment of LCs. As illustrated in Figure 8a, the softest tails with $k_{\theta }=10$ fail to align the mesogens far away from the interface, and the transition has to be shifted to a higher surface coverage of amphiphiles. However, this effect is too weak to be practical for nematic LCs, as indicated by the tiny shift shown in Figure 8b. In contrast, for the smectic-A phase at T=0.3 in Figure 8c, it is interesting to find that the discontinuous planar-to-homeotropic transition is gradually shifted to a higher surface coverage with the increasing tail rigidity of amphiphiles. These distinguishable shifts make smectic LCs promising candidates for sensing the subtle change in the chain rigidity of amphiphiles.

## 4. Conclusions

In summary, we have employed dissipative particle dynamics simulations in this study to explore unique anchoring behavior of different mesophases at amphiphile-laden aqueous-LC interfaces. By varying the surface coverage and tail rigidity of amphiphiles at different temperature, equilibrium anchoring configurations of nematic and smectic LCs are simulated and characterized to assess the sensitivity and selectivity of LC-based sensors as well as their underlying mechanisms. It is found that two distinct anchoring sequences, i.e., a continuous planar-tilted-homeotropic transition and a discontinuous planar-to-homeotropic transition with the increase of the amphiphile surface coverage, can be observed in nematic and smectic-A phases, respectively. More importantly, the latter occurs at a much lower surface coverage of amphiphiles, demonstrating a higher sensitivity for the smectic-based sensors. In addition, the reorientation dynamics further reveals that the interdigitation coupling between the mesogens and hydrophobic tails of amphiphiles plays a critical role of directing the tilted or homeotropic anchoring of nematic LCs in an interface-to-bulk propagation manner. In contrast, the homeotropic smectic anchoring is mainly governed by the appearance and growth of smectic layers through the LCs in a synchronous manner. Furthermore, it is also proven that the smectic LCs may possess a potential selectivity in response to a subtle change in the chain rigidity of amphiphiles.

These simulation findings are very interesting. It not only theoretically demonstrates the high sensitivity and versatile selectivity of the smectic-based sensors, but also provides a deeper understanding of the underlying mechanisms, which would be very valuable for the development and applications of novel LC-based sensors exploiting smectic or other mesophases.

## Figures and Tables

**Figure 1 molecules-27-07433-f001:**
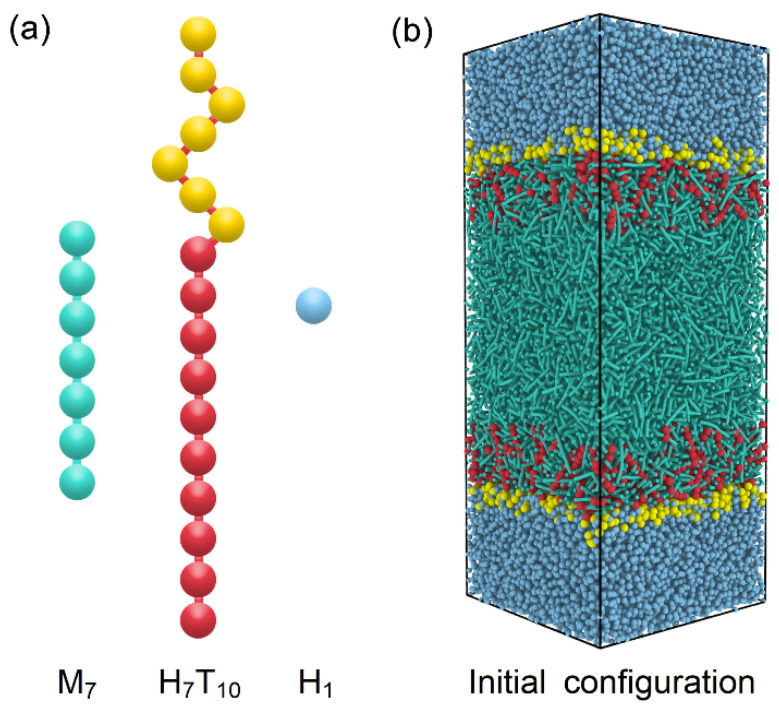
Schematic illustrations of (**a**) coarse-grained models of mesogens (M_7_), amphiphiles (H_7_T_10_) and water (H_1_) and (**b**) initial configuration of DPD simulations.

**Figure 2 molecules-27-07433-f002:**
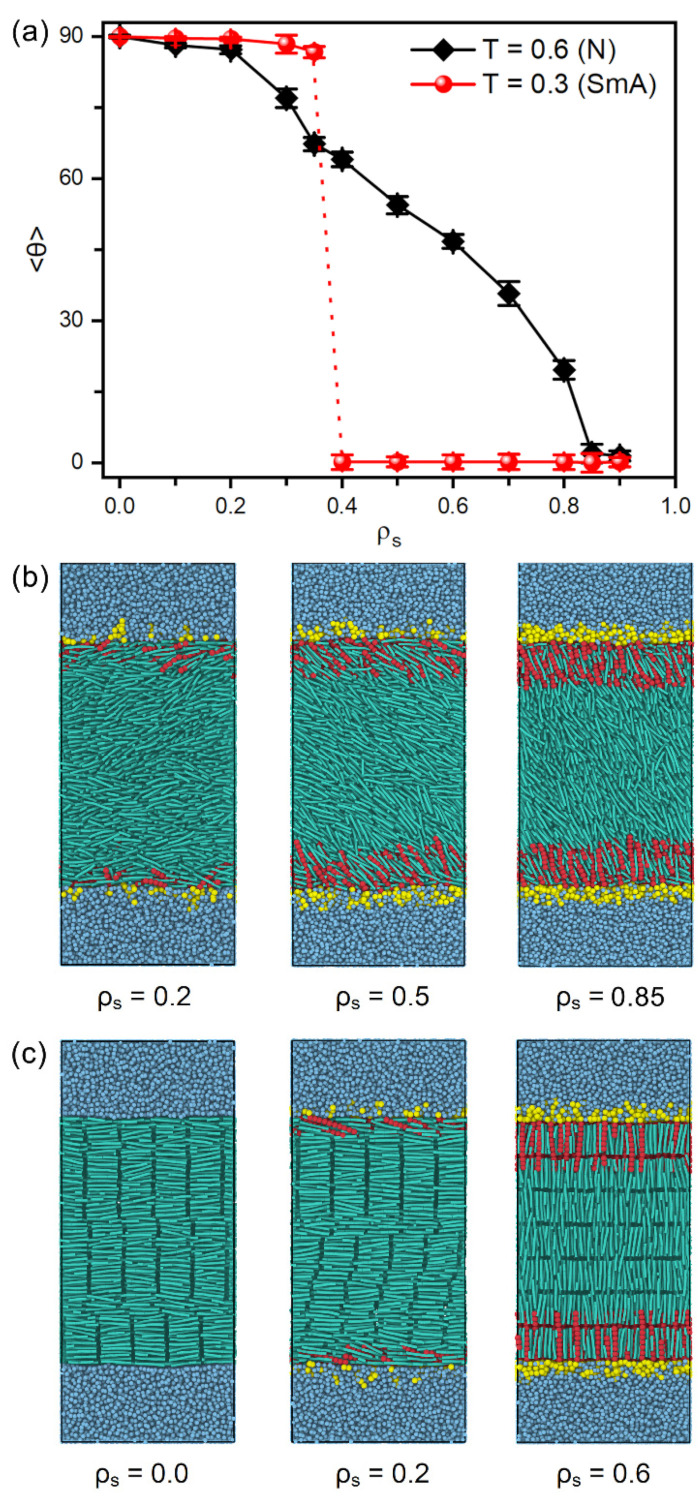
(**a**) Tilt angle of mesogens at the aqueous-LC interface as a function of the surface coverage of amphiphiles and the system temperature, and instantaneous snapshots of the representative anchoring configurations of (**b**) nematic (T=0.6, N) and (**c**) smectic (T=0.3, SmA) LCs.

**Figure 3 molecules-27-07433-f003:**
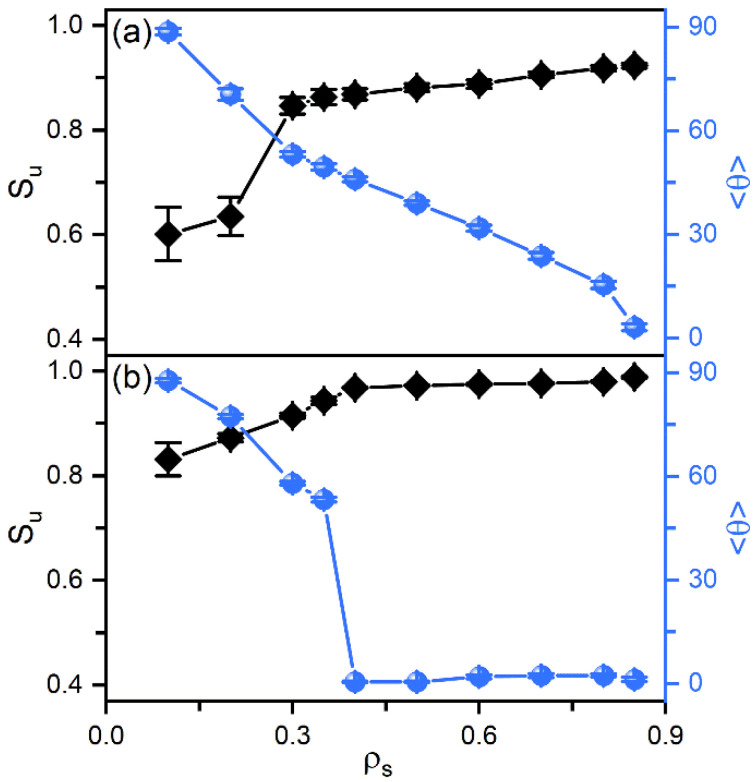
The orientational order parameter $S_{u}$ and tilt angle θ of the amphiphile tails as a function of the surface coverage in the anchoring transition sequences of (**a**) nematic (T=0.6) and (**b**) smectic (T=0.3) LCs.

**Figure 4 molecules-27-07433-f004:**
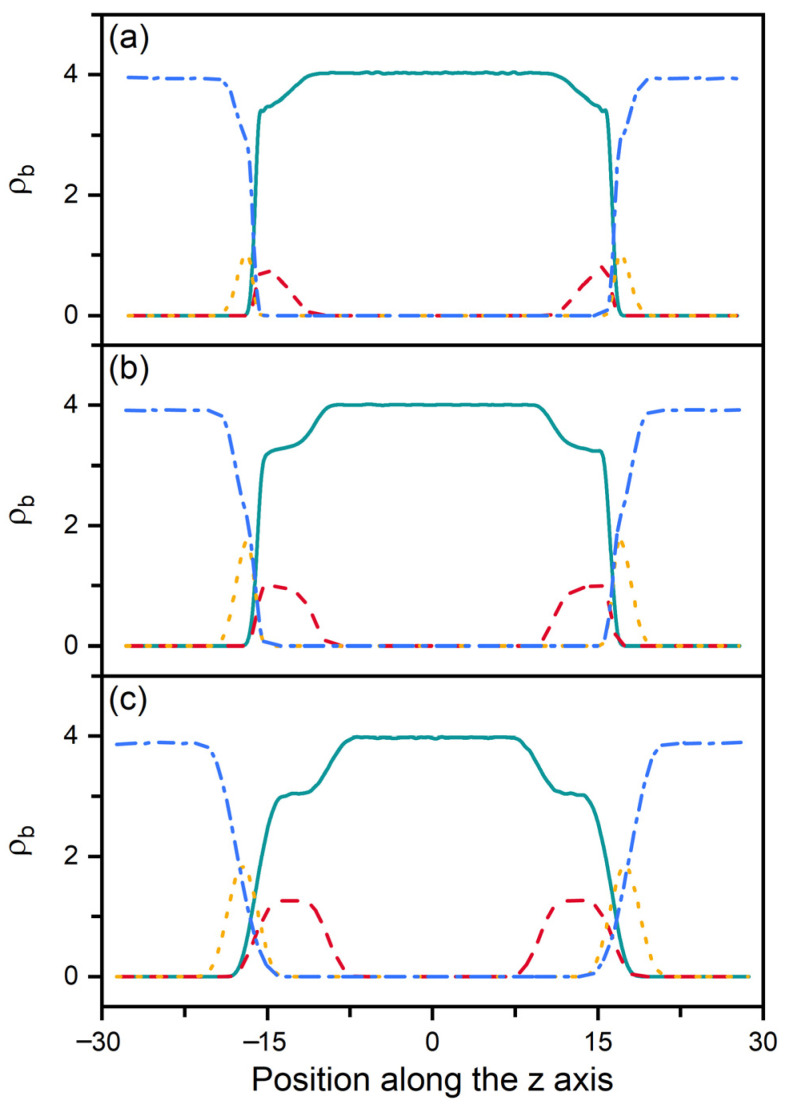
Bead density distributions of the mesogens (solid), amphiphile tails (dash), amphiphile heads (dot) and water (dash dot) in three representative anchoring configurations of nematic (T=0.6) LCs: (**a**) planar anchoring at $\rho_{s}$ = 0.2, (**b**) tilted anchoring at $\rho_{s}$ = 0.5 and (**c**) homeotropic anchoring at $\rho_{s}$ = 0.85.

**Figure 5 molecules-27-07433-f005:**
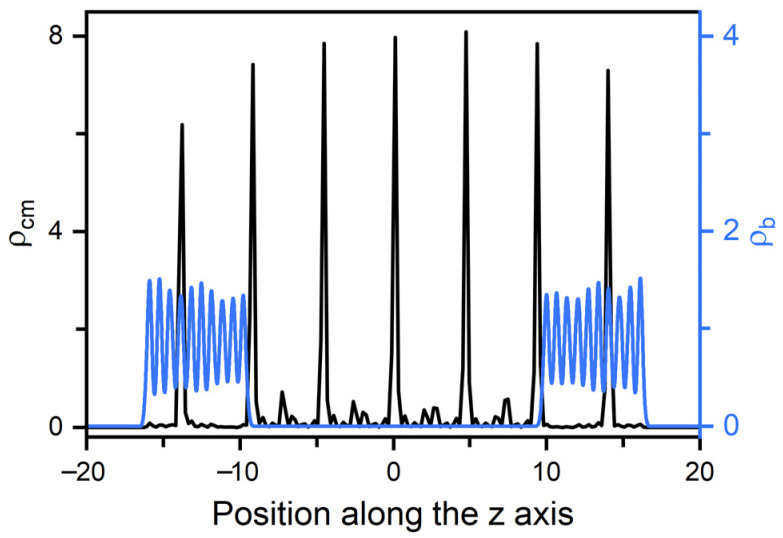
Density distributions for the center of mass of mesogens and the beads of amphiphile tails in a typical homeotropic smectic anchoring at T=0.3 and $\rho_{s}$ = 0.6. The corresponding bead density distributions are presented in Appendix A.

**Figure 6 molecules-27-07433-f006:**
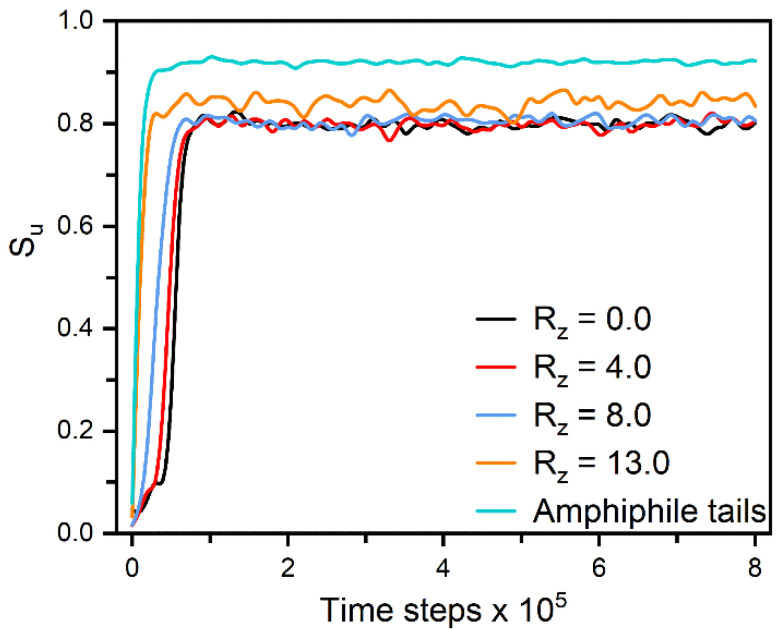
Time evolution of the orientational order parameter for the amphiphile tails and mesogens in slabs along the z axis when directly cooling a disordered configuration with $\rho_{s}$ = 0.85 to the homeotropic nematic anchoring at T=0.6. Each slab with a center locating at $R_{z}$ has a thickness of 4.0, and the slab with $R_{z}$ = 0.0 is just in the center of bulk LCs. Instantaneous snapshots of six representative configurations in this ordering process are displayed in Appendix A.

**Figure 7 molecules-27-07433-f007:**
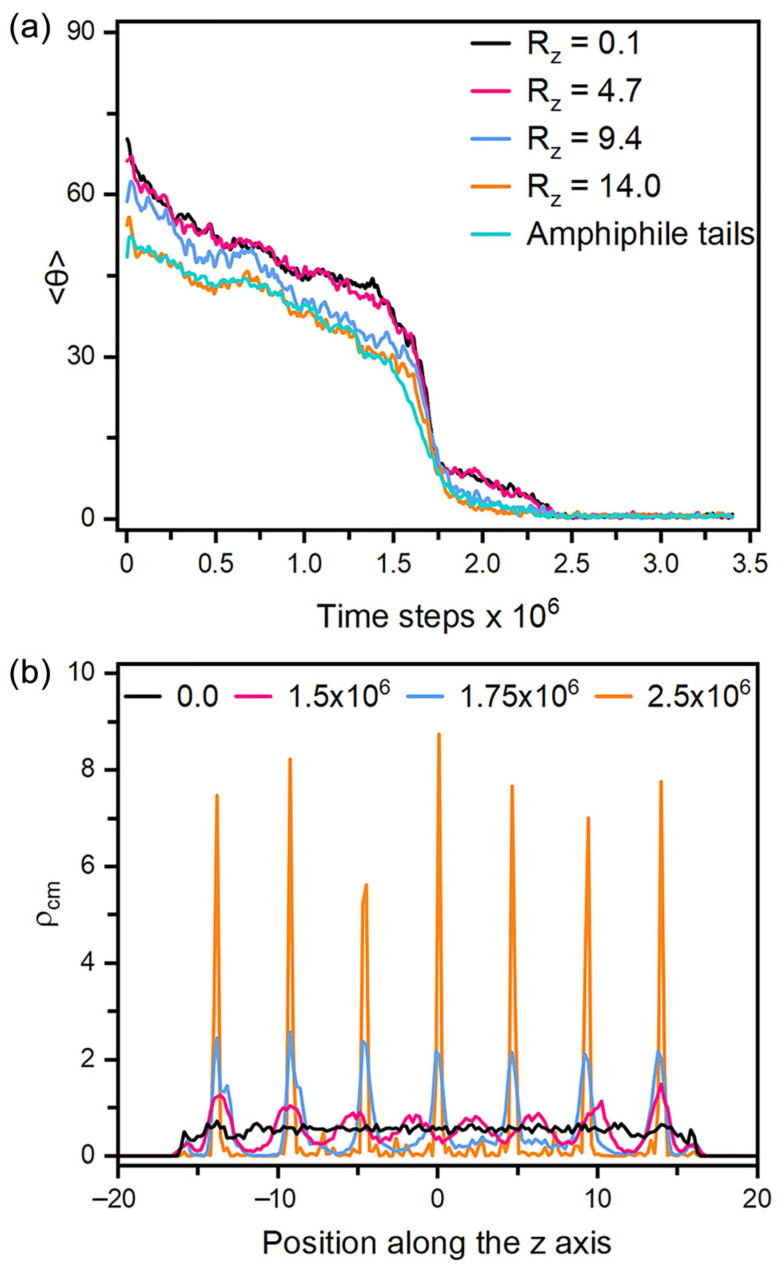
(**a**) Time evolution of the tilt angle for the mesogens in each smectic layer along the z axis and the amphiphile tails when directly cooling a tilted configuration with $\rho_{s}$ = 0.4 and T=0.6 to the homeotropic smectic anchoring at T=0.3, and (**b**) the density distributions for the center of mass of mesogens in four instantaneous configurations of the above process at t = 0.0, 1.5 × 10^6^, 1.75 × 10^6^ and 2.5 × 10^6^ time steps. Instantaneous snapshots of six representative configurations in this reorientation process are displayed in Appendix A.

**Figure 8 molecules-27-07433-f008:**
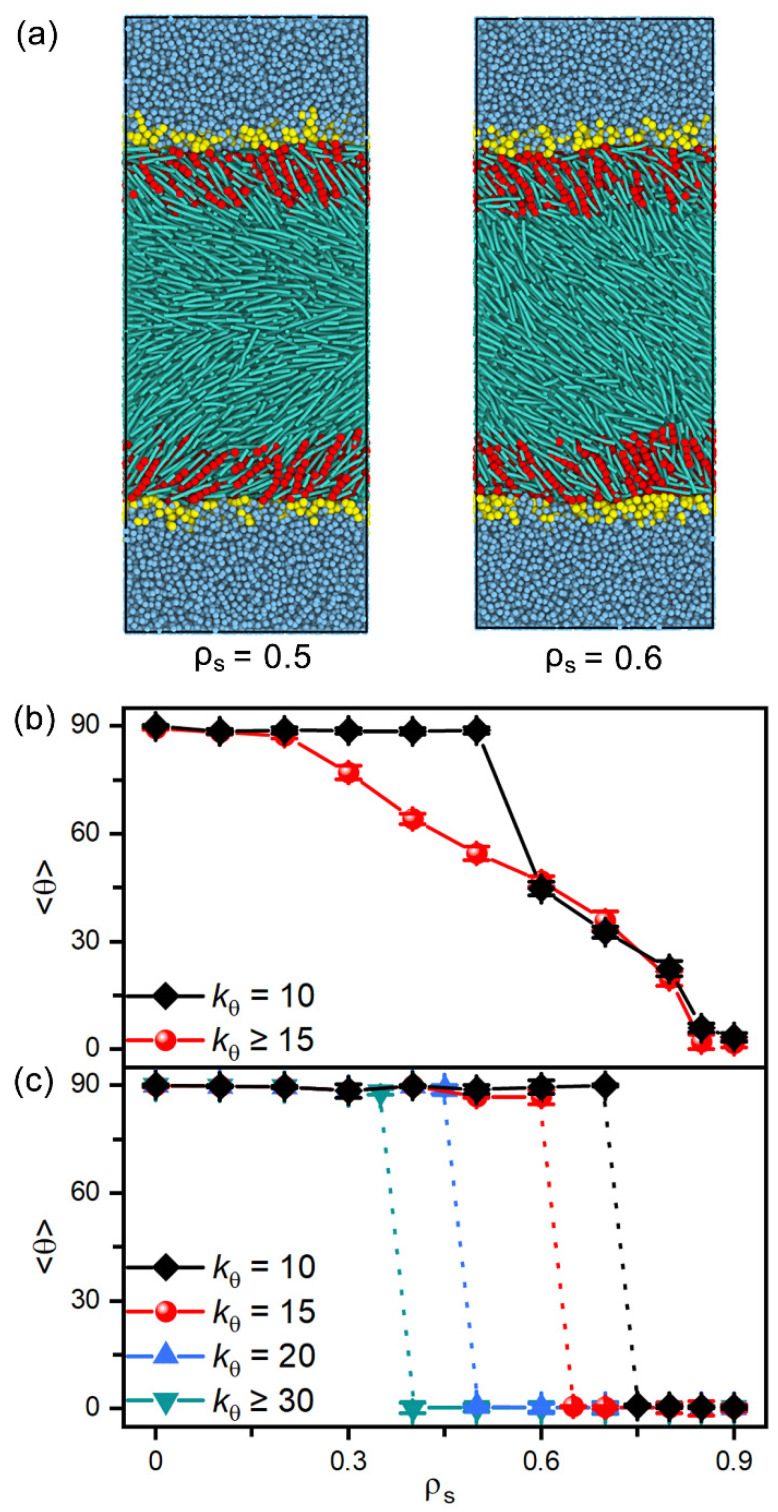
(**a**) Snapshots of the representative planar and tilted anchoring configurations induced by the amphiphiles with softest tails of $k_{\theta }$ = 10. The tilt angle of (**b**) nematic and (**c**) smectic LCs as a function of the surface coverage of amphiphiles with tunable tail rigidity of $k_{\theta }$ = 10, 15, 20, 30 and 50.

## Data Availability

All the data are presented in the manuscript.

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
