# Peer review of "Dissipative Particle Dynamics Simulation of the Sensitive Anchoring Behavior of Smectic Liquid Crystals at Aqueous Phase"

_molecules, 2022, doi:10.3390/molecules27217433_

Round 1

Reviewer 1 Report

Review of manuscript

Number: molecules-1971836

Title: Dissipative particle dynamics simulation of the sensitive anchoring behavior of smectic liquid crystals at aqueous phase

The manuscript employs dissipative particle dynamics to describe the unique anchoring behavior of nematic and smectic liquid crystal-based sensors at amphiphile-laden aqueous-LC interface. In this case, simple generic DPD model that contains rid-like models of LCs is used. To evidence the unique behavior the authors measured orientational order parameter, tilt angle and density profiles of mesogens, amphiphiles in the system together with density profiles of surrounding liquid (water in that case). The authors showed that orientation of amphiphiles on the water/mesogen interface influence the orientation of mesogens in their bulk phase

Remarks

Line 82 – I think that for is extra in that sentence and should be deleted.

Line 127 – fine mapping scheme is used for the model, to which LCs the model fits with its configuration? I would mention some real system

Line 140 – please add the range of Lz values

Simulation Setup – If I understood well, you have used classical NVT ensemble for modelling the interface.

Simulation Setup – Did you use some existing simulation package or your code? Please add the info.

No correlations are measured to obtain the speed of equilibrium. Autocorrelation function time can give proper estimation of the equilibration period and can serve for distinguishing the systems from each other or even its components.

I am missing some benchmarks for system size scaling or discussion of the Lz length if it is comparable with the experimental systems (once the CG is set by one water molecule in one DPD bead)

Figure 2a, Figure 8c – If the process is discontinuous in nature (see the text in line 185) the lines should not be continuous.

Figure 4 – I am missing same figures for other phases, otherwise I cannot compare directly the results.

Line 281 – I do not understand the sentence: However, the mismatch between the spacing of smectic layers and the thickness of LC films indeed significantly affects the alignment of smectic LCs in our simulations – The size of the mesogens are much smaller that the size of simulation cell, e.g. there should be no mismatch. Or If I am wrong, please specify clearly what do you mean by that. I also think that increasing the length of Lz will affect the ordering in the bulk phase.

Lines 3223-332 – missing the message for the experimentalists, need to be emphasized more.

Questions

·       What is the real size of the simulation box compared to your coarse-graining level which is rc=0.44 nm? Does it fit the cell in real LCs?

If relevant: What is the size Lz when the mesogens loose the orientation of their counterparts close to the interface? It seems to me that (from Figure 5) ordering in the middle of the box is slightly disturbed and I am curios how long the ordering holds.

What is the speed of ordering of mesogens in the simulation compared to different conditions (size, phase, transition between phases etc..)? Please express in terms of correlation functions.

Why you did not use the NsigmaT ensemble where the constant surface tension is used. Did not you have a problem with interface stability during the simulation?

Can you provide the SI with additional info, at least about ordering and reordering dynamics?

Decision

According to above mentions remarks and questions I think that major revision is needed for this manuscript. Mainly, I would add the correlation function and correlation time estimation. Moreover, I would appreciate the information about box size influence on ordering and the range of ordering. Finally, I would add the information about real dimensions that can connect the generic model with real systems. I think that adding this information can make the manuscript more attractive for broader audience including experimental section.

Reviewer 2 Report

The presented paper is devoted to the solution of practically important problem concerning the usage of liquid crystals (LC) as sensors of biological impurities. The authors made numerical simulations which may be useful to clarify the important problem of selectivity of LC biosensors, which play the key role in practical application. The details of the numerical simulation are described in details and obtained results are correct in the framework of used approximation. There is one question concerning the initial state of the system under simulation. Namely, what changes in the results of calculation will take place if in the initial state LC molecules of will form the orientational (for nematics) and translational (for smectics) order? Such initial state is more convenient for practical usage of LC sensors.

In general, the paper may be recommended for publication without changes .

Round 2

Reviewer 1 Report

The authors carefully answered all my questions and worked on my comments. Some of these were inserted into the manuscript and also into the SI. Therefore, I agree to publish the paper in its present form.